# CA125 as a Marker of Heart Failure in the Older Women: A Population-Based Analysis

**DOI:** 10.3390/jcm8050607

**Published:** 2019-05-03

**Authors:** Weronika Bulska-Będkowska, Elżbieta Chełmecka, Aleksander J. Owczarek, Katarzyna Mizia-Stec, Andrzej Witek, Aleksandra Szybalska, Tomasz Grodzicki, Magdalena Olszanecka-Glinianowicz, Jerzy Chudek

**Affiliations:** 1Department of Internal Diseases and Oncological Chemotherapy, School of Medicine in Katowice, Medical University of Silesia, 40-027 Katowice, Poland; chj@poczta.fm; 2Department of Statistics, Department of Instrumental Analysis, School of Pharmacy and Laboratory Medicine in Sosnowiec, Medical University of Silesia, 41-200 Sosnowiec, Poland; echelmecka@sum.edu.pl (E.C.); aowczarek@paintbox.com.pl (A.J.O.); 3First Department of Cardiology, School of Medicine in Katowice, Medical University of Silesia, 40-635 Katowice, Poland; kmizia@op.pl; 4Department of Gynecology and Obstetrics, School of Medicine in Katowice, Medical University of Silesia, 40-752 Katowice, Poland; awitek@sum.edu.pl; 5International Institute of Molecular and Cell Biology, Warsaw 02-109, Poland; a.szybalska@iimcb.gov.pl; 6Department of Internal Medicine and Gerontology, Jagiellonian University Medical College, 31-531 Cracow, Poland; tomekg@su.krakow.pl; 7Health Promotion and Obesity Management Unit, Department of Pathophysiology, School of Medicine in Katowice, Medical University of Silesia, 40-752 Katowice, Poland; magols@esculap.pl; 8Pathophysiology Unit, Department of Pathophysiology, School of Medicine in Katowice, Medical University of Silesia, 40-752 Katowice, Poland

**Keywords:** carbohydrate antigen-125, heart failure, inflammatory marker, older women

## Abstract

(1) Background: Cancer antigen 125 (CA125) is a glycoprotein that is expressed by tissue derived from coelomic epithelium in the pleura, peritoneum, pericardium. It has been shown that CA125 concentrations are correlated with NT-proBNP in older people with congestive heart failure (HF). We conducted a study on the association between concentrations of CA125 and NT-proBNP in a population-based cohort of older Polish women. (2) Methods: The current research is sub-study of a large, cross-sectional research project (PolSenior). The study group consisted of 1565 Caucasian women aged 65–102 years. To assess the relationship between CA125 and other variables a stepwise backward multivariate normal and skew-t regression analyses were performed. (3) Results: The median of CA125 concentration was 13.0 U/mL and values over the upper normal range limit (35 U/mL) were observed in 5.1% (*n* = 79) of the study cohort. The concentration of CA125 was positively related to age, hospitalization for HF and history of atrial fibrillation and chronic obstructive pulmonary disease, levels of NT-proBNP, IL-6, hs-CRP and triglycerides. We found in the multivariate analyses, that increased CA125 levels were independently associated with log_10_ (IL-6) (β = 11.022), history of hospitalization for HF (β = 4.619), log_10_ (NT-proBNP) (β = 4.416) and age (β = 3.93 for 10 years). (4) Conclusions: Despite the association between CA125 and NT-proBNP, the usefulness of CA125 for the detection of HF in older women is limited by factors such as inflammatory status and age.

## 1. Introduction

Cancer antigen 125 (CA125) is a glycoprotein that is expressed by tissue derived from coelomic epithelium and has a molecular weight estimated to range from 110 to more than 2000 kD [1]. The CA125 serum level has been shown to be increased in women with ovarian cancer and less often in breast [2], lung [3] and gastrointestinal cancers [4]. However, it is not a screening test for malignancies as it may also be elevated in a variety of benign conditions (such as pregnancy, endometriosis, uterine leiomyomata, pelvic inflammatory disease, cirrhosis, peritonitis, pleuritis, pancreatitis, and tuberculosis) [5].

Several recent studies have shown that serum CA125 levels are also elevated in heart failure (HF), atrial fibrillation (AF), and ischemic heart disease [6,7,8]. Only a few studies that have assessed the CA125 levels in selected groups of patients after cardiac surgery (cardiac transplantation, transcatheter aortic valve implantation) have been published so far [9,10]. It seems that neurohormonal and inflammatory activation, as well as increased central venous volume and congestion, are factors that increase the level of CA125 [11,12]. 

It is worth noting that the serum concentration of CA125 was associated with the severity of clinical conditions and poor short-term prognosis in patients with cardiovascular disease [13]. Ma et al. [14] showed a significant correlation between serum concentration of CA125 and the clinical status of patients aged 85 and older with congestive HF as well as the usefulness of CA125 as a prognostic factor of death and rehospitalization. Moreover, Yucel et al. [15] reported that the serum concentration of CA125 may be used to predict AF in patients with systolic HF. In addition, a significant correlation between CA125 and NT-proBNP levels was found in Chinese older patients only with congestive HF. Moreover, it should be noted that this group included only less than 16% of women [14]. It is well known that the clinical course of cardiovascular diseases differs between men and women [16], thus the separate analysis of biomarkers and risk factors in men and women seems reasonable. Furthermore, the impact of chronic diseases and inflammatory markers on CA125 levels in a population-based cohort has not been studied, yet.

The main aim of this study was to assess the relationships between the serum concentration of CA125 and NT-proBNP in a population-based cohort of older Polish women. 

## 2. Materials and Methods

The current study was part of the PolSenior project, which was a large, cross-sectional, multicenter, interdisciplinary research project performed among the older Polish adult population (4979 respondents aged 65 and older, including 2412 women). Only women aged 65 and older with an assessed serum level of CA125 (*n* = 1951) were included in this sub-study. The exclusion criteria were as follow: (1) history of neoplastic disease (*n* = 125), (2) hepatitis B virus or hepatitis C virus infection (*n* = 52), (3) lack of information about the cardiovascular disease (*n* = 158), (4) lack of NT-pro BNP assessment (*n* = 51). Finally, the study group consisted of 1565 older Polish women (Figure A1). 

The PolSenior project consisted of three stages: (1) conducting a health and socioeconomic survey by nurses trained for this purpose nurses that included comprehensive geriatric assessment, (2) measurements of body mass, height, waist circumference and blood pressure, (3) the collection of blood and urine samples. 

The study was approved by the Bioethics Committee of the Medical University of Silesia in Katowice (KNW/0022/KB1/38/II/08/10; KNW-6501-38/I/08). Before enrollment, written and informed consent was obtained from all subjects or their caregivers.

### 2.1. Biochemical Measurements

Serum concentrations of CA125 and NT-proBNP were measured by the electrochemiluminescence method (ECLIA) using an Elecsys 2010 (CA125) and Cobas E411 (NT-proBNP) analyzers (Roche Diagnostics GmbH, Mannheim, Germany), with an intermediate precision of <4.2%, and 4.6%, respectively. The sensitivity of the method for CA125 was 0.6 U/mL. According to the current guidelines, a value of NT-proBNP below 125 pg/mL was considered as the exclusion criteria for the diagnosis of heart failure [17].

Serum levels of C-reactive protein were assessed by a the latex-enhanced immunoturbidimetric assay using an automated system (Modular PPE, Roche Diagnostics GmbH, Mannheim, Germany) with a limit of quantification (LoQ) of 0.11 mg/L and intermediate precision of <5.7%. The biochemical tests such as total cholesterol, high-density lipoprotein (HDL), low-density lipoprotein (LDL), triglycerides, creatinine (Jaffa method) were also measured using an automated system (Modular PPE) with intermediate precisions of <1.7%, <1.3%, 1.2%, <1.8% and <2.3%, respectively. 

Urinalysis by the Combur-Test (Roche Diagnostics, Mannheim, Germany) was performed in all urine samples using the Miditron M system (Roche Diagnostics, Mannheim, Germany). Albuminuria was diagnosed if the albumin concentration in the urine was >30 mg/L. If albuminuria was not detected in the urine strip test, the albumin concentration was measured by the immunoturbidimetric method (Roche Diagnostics, Mannheim Germany) with high sensitivity (LoQ of 3 mg/L). 

Plasma interleukin 6 (IL-6) was assessed by ELISA using a kit produced by R and D Systems (Minneapolis, MN, USA) with a LoQ of 0.11 pg/mL and intermediate precision of <6.5%. Serum insulin levels were assessed by the electrochemiluminescence method (ECLIA) using commercially available kits on a Cobas E411 analyzer (Roche Diagnostics GmbH, Mannheim, Germany) with an intermediate precision of <3.8%. 

### 2.2. Data Analysis

Hospitalization for HF and/or myocardial infarction, diagnosis of AF and/or coronary artery disease and/or chronic obstructive pulmonary disease (COPD), currently applied treatment and smoking status were collected from the questionnaire survey. 

Diagnosis of diabetes was based on medical history, medication use, and fasting serum glucose above 125 mg/dL. Participants were considered to have hypertension if they had a mean systolic blood pressure (SBP) ≥140 mmHg and/or diastolic blood pressure (DBP) ≥90 mmHg or used antihypertensive medications [18]. 

The body mass index (BMI) was calculated as the weight (kg) divided by the square of the height (meters).

Estimated glomerular filtration rate (eGFR; mL/min/1.73 m^2^) was calculated according to the short MDRD (modification of diet in renal disease) formula. 

The albumin-to-creatinine ratio (ACR; mg/g) was calculated as the urine albumin concentration divided by the urine creatinine concentration. 

The homeostatic model assessment (HOMA-IR) was used to evaluate insulin resistance (fasting serum insulin (μU/mL) × fasting plasma glucose (mmol/L)/22.5). Insulin resistance was diagnosed if HOMA-IR was 2.5 or higher.

### 2.3. Statistical Analysis

Statistical analyses were performed using STATISTICA 10.0 PL (TIBCO Software Inc, Palo Alto, CA, USA) and StataSE 12.0 (StataCorp LP, TX, USA). Statistical significance was set at a *p*-value below 0.05. All tests were two-tailed. Nominal and ordinal data were expressed as percentages, while interval data were expressed as a mean value ± standard deviation in the case of a normal distribution, or as median (lower quartile–upper quartile) in the event of data with a skewed or non-normal distribution. To assess the relationship between CA125 and other variables, a stepwise backward multivariate normal and skew-t regression analyses were used, due to the heavily skewed distribution of some variables. The range of serum CA125 concentration was shown with the histogram and was modelled with the kernel density plot with the Epanechnikov kernel function (Figure A2).

## 3. Results

### 3.1. Characteristics of the Study Population

Our sub-study consisted of 1565 Polish women aged between 65–102 years. The average age was 78 ± 9 years. Characteristics of the study population according to age, medical conditions, biochemical parameters and medication used are shown in Table 1. 

The study cohort included: 1218 (78.2%) women diagnosed with hypertension, 291 (18.6%) with coronary artery disease, 160 (10.2%) previously hospitalized for heart failure, 105 (6.7%) with past myocardial infarction and 297 (19%) with a history of atrial fibrillation. 

As a consequence of diagnosed cardiovascular diseases, a large percentage of study participants were taking medications, including angiotensin-converting-enzyme inhibitors (ACE-I) and angiotensin II receptor blockers (ARB) (*n* = 764; 48.8%), β-blockers (*n* = 603; 38.5%), diuretics (*n* = 513; 32.8%), and the mineralocorticoid receptor blocker–spironolactone (*n* = 193; 12.3%). Every fourth woman received lipid-lowering drugs, mostly statins (*n* = 382; 24.4%) and very rarely fibrates (*n* = 19; 1.2%).

### 3.2. Serum CA125 Concentration

The median of CA125 serum concentration was 13.0 U/mL (lower and upper quartile: 9.72–17.60, range: 1.1–225.9 U/mL). CA125 levels over the upper normal range limit (35 U/mL) were found in 79 women (5.1%, see Figure A2).

In a univariate analysis serum concentration of CA125 was positively related to age, hospitalization for HF and history of AF and COPD as well as the serum levels of NT-proBNP, IL-6, hsCRP, and triglycerides (Figure 1). The CA125 concentration was inversely related to BMI, the concentration of HDL-cholesterol and eGFR values (Table 2). There was no association between CA125 concentration and the occurrence of diabetes, hypertension, coronary heart disease and history of myocardial infarction as well as used cardiac medication (Table 2).

To assess the factors affecting CA125 serum concentration, a multivariate stepwise backward skew-t regression model was used with independent factors selected based on univariate analyses, and the results are shown in Table 2. We found that increased concentration of CA125 was independently associated with log_10_ (Il-6) (β = 11.022), history of hospitalization for HF (β = 4.619), log_10_ (NT-proBNP) (β = 4.416) and age (β = 3,93 for 10 years), as shown in Figure 2.

## 4. Discussion

To the best of our knowledge, the current study was the first to investigate the association between CA125 and NT-proBNP levels in a population-based cohort of older Caucasian women. Ma et al. [14] conducted a similar study, but among Chinese patients hospitalized for chronic HF aged 85 years and older, with the majority of the study group being men (84%). It should be noted that race is an independent factor influencing CA125 levels. Pauler et al. [19] showed that CA125 levels were the highest in Caucasians, lower in Asians and the lowest in African healthy postmenopausal women.

We found that the serum concentration of CA125 was independently associated with NT-proBNP. This association may suggest a similar release mechanism related to an elevation in intracavitary pressures, venous pressure and stress of cardiac walls [20]. 

Nunez et al. [21] showed that the simultaneous increase in the values of CA125 and NT-proBNP was associated with the highest risk of mortality due to acute HF and that the increase of the value of one marker was also related to the indirect risk. Mendez et al. [22] suggested that the value of CA125 over 60 U/L (higher than the upper range of normal values) may identify patients in chronic HF with poor outcome. However, there are insufficient studies to determine the limit of CA125 value for the assessment of mortality risk in patients with HF. On the basis of our results, the assessment of CA125 usefulness as a marker of HF severity and mortality for HF is not possible. The association between CA125 and NT-proBNP levels is not enough to prove its’ usefulness in the work-up of HF. Nevertheless, the findings of other researchers that have shown an association between CA125 and New York Heart Association (NYHA) functional classification scale [13,23,24,25], ejection fraction of the left ventricle [15,26,27], systolic pulmonary artery pressure [13,15,25,28,29], pulmonary artery wedge pressure [13], right atrial pressure [13], left atrial volume index [30], are interesting and stimulate for further studies.

The obtained results also showed that serum CA125 levels were independently associated with the history of hospitalization for HF and increased levels of IL-6. This is in accordance with studies that have shown that the increased CA125 concentration in HF is associated with mechanical stress and systemic inflammation [31]. In addition, it has been shown that inflammatory cytokines activate CA125 synthesis by the mesothelial cells [32]. In HF, high venous pressure can lead to the congestion and increased hydrostatic pressure on the mesothelium [33], stimulating the release of inflammatory markers such as IL-6 or IL-10, TNF-α [11,12]. We have found a statistical association between CA125 and IL-6, supporting the hypothesis that elevated levels of IL-6 in cardiovascular diseases may play a leading role in the stimulation of CA125-producing cells in a damaged mesothelium. Other studies have also confirmed the association between cytokines and CA125 levels in subjects with HF [12,34]. 

In the present study, the serum levels of CA125 were positively related to the history of atrial fibrillation (AF), but we did not demonstrate its independent association in a multivariate analysis. AF was found in up to 30% of subjects with HF [28], therefore we supposed that AF may have a secondary role in the increase of CA125 values. In addition, Sekiguchi et al. [35] revealed that a high CA125 concentration was an independent predictor of new-onset AF in healthy postmenopausal women without HF. It has also been shown that increased levels of cytokines such as IL-2, IL-6, IL-8, TNF-α in subjects with new-onset AF, can stimulate mesothelial cells to produce CA125 [36]. Recently published studies have demonstrated the independent effect of permanent AF on CA125 levels [28]. Despite the observed results in our study of an association between CA125 and IL-6, we could not verify whether CA125 is a predictor of AF in the course of HF. 

In addition, we did not find an independent association between the occurrence of coronary heart disease and a history of myocardial infarction and CA125 serum concentration. This is in contrast to the results obtained by Yalta et al. [37], which revealed increased CA125 concentration in patients with acute myocardial infarction within 72 h of the incident. It is possible that the damage to the myocardium caused a decline in myocardial performance with an increase in central venous pressure in some patients and inflammatory response that stimulated mesothelial cells. 

We also found an association between the occurrence of chronic obstructive pulmonary disease and CA125 concentration. Other studies have also confirmed a relationship between COPD and CA125 levels [38,39,40]. Chronic obstructive pulmonary disease is characterized by abnormal enlargement of the right ventricular (RV), leading to heart failure. Yilmez et al. [41] suggested that high CA125 levels were associated with RV failure. The high concentrations of CA125 in COPD are probably caused by increased systolic pulmonary artery pressure, which is considered equal to the RV systolic pressure. However, our data cannot support this observation. 

Finally, this study demonstrated the independent association of CA125 concentration with age. Our study showed a positive association between age and the serum levels of CA125. This study was the first one to analyze a large group of Caucasian women with a wide age span (65–102 years). In a previous study, Johnson et al. [42] observed similar results among multi-ethnic women without cardiac disease aged between 55–74 years. Pauler et al. [19] demonstrated the opposite results, where a decrease in CA125 concentrations with aging in postmenopausal women during the 12-year follow-up period. However, the study was carried out among a group of younger and healthy women aged between 40–60 years. In addition, Sikaris et al. [43] described that the CA125 concentration gradually increased after the age of 70 years, both in men and in women. 

In summary, our study showed the relationship between CA 125 and NT-proBNP levels. In addition, we observed that increased CA125 levels may be related to numerous factors such as age, hospitalization for HF, history of AF and COPD, as well as increased levels of inflammatory markers (IL-6, hs-CRP). Thus, the clinical usefulness of CA125 as a marker of HF diagnosis and severity is limited. The usefulness of CA125 as a prognostic marker for HF severity and mortality in Caucasian women population requires further study. 

This study has some limitations, mostly related to the lack of objective measures of the severity of HF. We did not evaluate echocardiographic metrics as it is difficult to apply in population-based studies performed in the participants’ place of living. Hospital discharge cards shown by participants do not exclude the occurrence of compensated HF and past asymptomatic myocardial infarction in some participants. Moreover, CA125 was not assessed in men. This is why the final results cannot be generalized for the whole population aged over 65 years. 

The strength of this study was the inclusion of the oldest women representative of the Polish population. The study group included women aged between 65 and 102, which allowed the conclusion that age affects the value of the marker. Moreover, we were the first to examine the association between CA125 and NT-proBNP levels in a large population-based cohort of Caucasian women and assess the factors affecting the circulating CA125 levels.

## 5. Conclusions

Despite the association between CA125 and NT-proBNP, the usefulness of CA125 for the detection of HF in the older women is limited by factors such as inflammatory status and age.

## Figures and Tables

**Figure 1 jcm-08-00607-f001:**
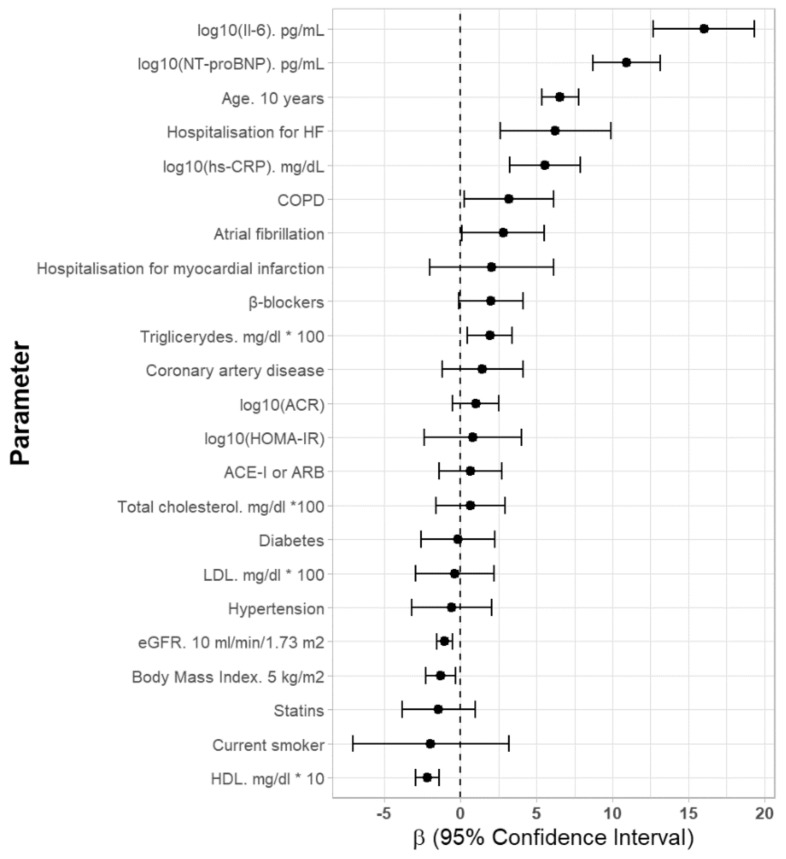
Univariate parameters associated with serum CA125 concentrations in older Polish women. Abbreviations: Il-6—interleukin 6, HF—heart failure, hs-CRP—high sensitivity C-reactive protein, COPD—chronic obstructive pulmonary disease, ACR—albumin-to-creatinine ratio, HOMAR-IR—homeostatic model assessment, ACE-I—angiotensin-converting-enzyme inhibitor, ARB—angiotensin receptor blockers.

**Figure 2 jcm-08-00607-f002:**
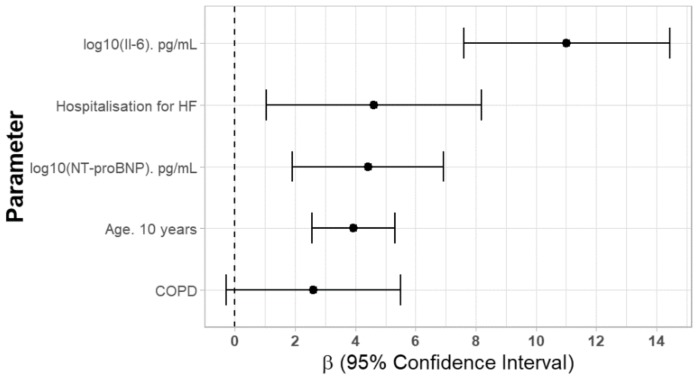
Independent factors affecting serum CA125 concentrations. Abbreviations: Il-6—interleukin 6, HF—heart failure, COPD—chronic obstructive pulmonary disease.

**Table 1 jcm-08-00607-t001:** Characteristics of the study population according to age, medical conditions, biochemical parameters and medication used. Data are provided as mean ± standard deviation (SD ) , median (1–3 Q) or numbers (percentage).

	All (*n* = 1565)
Age (years)	78 ± 9
65–69 years, *n* (%)	293 (18.7)
70–74 years, *n* (%)	321 (20.5)
75–79 years, *n* (%)	259 (16.5)
80–84 years, *n* (%)	229 (14.6)
85–90 years, *n* (%)	233 (14.9)
>90 years, *n* (%)	230 (14.7)
BMI (kg/m^2^)	29.1 ± 5.5
<18.5 kg/m^2^, *n* (%)	16 (1.1)
18.5–24.9 kg/m^2^, *n* (%)	324 (2.2)
25–29.9 kg/m^2^, *n* (%)	527 (36.2)
≥30 kg/m^2^, *n* (%)	590 (40.5)
Current smoker, *n* (%)	67 (4.3)
Hypertension, *n* (%)	1218 (78.2)
Coronary artery disease, *n* (%)	291 (18.6)
Hospitalization for MI, *n* (%)	105 (6.7)
Hospitalization for HF, *n* (%)	160 (10.2)
AF, *n* (%)	297 (19.0)
Diabetes, *n* (%)	384 (24.5)
COPD, *n* (%)	249 (15.9)
Glucose, mg/dL	94.5 (85.7–107.1)
Impaired fasting glucose, *n* (%)	558 (35.6)
HOMA-IR	1.44 (0.74–2.76)
Total cholesterol, mg/dL	211.1 ± 47.2
LDL-cholesterol, mg/dL	124.9 ± 41.3
HDL-cholesterol, mg/dL	52.9 ± 13.8
Triglycerydes, mg/dL	121.5 (93.1–158.9)
eGFR-MDRD short, mL/min/1.73 m^2^	73.5 ± 21.1
eGFR < 60 mL/min/1.73 m^2^, *n* (%)	387 (24.7)
ACR	5.56 (2.36–15.75)
ACR ≥ 300, *n* (%)	39 (2.7)
hs-CRP, mg/dL	2.47 (1.21–4.88)
Il-6, pg/mL	2.2 (1.5–3.7)
NT-proBNP, pg/mL	225 (114–504)
Ca125, U/mL	13.0 (9.7–17.6)
Ca125 > 35 U/mL, *n* (%)	79 (5.1)
β-blockers, *n* (%)	603; 38.5%
ACE-I or ARB, *n* (%)	764 (48.8)
Diuretics, *n* (%)	513 (32.8)
Spironolactone, *n* (%)	193 (12.3)
Statins, *n* (%)	382 (24.4)
Fibrates, *n* (%)	19 (1.2)

Abbreviations: BMI—body mass index, MI—myocardial infarction, HT—heart failure, AF—atrial fibrillation, COPD—chronic obstructive pulmonary disease, HOMAR-IR—homeostatic model assessment, ACR—albumin-to-creatinine ratio, hsCRP—high-sensitivity C-reactive protein, Il-6—interleukin 6, ACE-I—angiotensin-converting-enzyme inhibitor, ARB—angiotensin receptor blockers. Data provided as mean ± standard deviation (SD), median (1–3 Q) or numbers (percentage).

**Table 2 jcm-08-00607-t002:** Univariate and multivariate, stepwise, backward skew-t regression analyses of factors associated with increased CA125 serum concentrations in older Polish women.

	Univariate			Multivariate		
log_10_ (CA125 U/mL) * 100	β	95% CI: β	*p*	β	95% CI: β	*p*
Age (years)	0.656	0.534–0.777	<0.001	0.393	0.255–0.532	<0.001
BMI (kg/m^2^)	−0.260	−0.454–−0.066	<0.01	-	-	-
Current smoker (Yes)	−1.950	−7.107–3.206	0.46	-	-	-
AF (Yes)	2.805	0.081–5.528	<0.05	-	-	-
Coronary artery disease (Yes)	1.438	−1.222–4.098	0.29	-	-	-
Hospitalization for MI (Yes)	2.040	−2.043–6.124	0.33	-	-	-
Hospitalization for HF (Yes)	6.249	2.615–9.882	<0.01	4.619	1.042–8.196	<0.05
Hypertension (Yes)	−0.577	−3.218–2.065	0.67	-	-	-
Diabetes (Yes)	−0.183	−2.603–2.237	0.88	-	-	-
COPD (Yes)	3.173	0.234–6.111	<0.05	2.607	−0.281–5.495	0.08
Total cholesterol (mg/dL) *100	0.636	−1.635–0.291	0.58	-	-	-
LDL-cholesterol (mg/dL) * 100	−0.366	−2.931–2.198	0.78	-	-	-
HDL-cholesterol (mg/dL)	−0.218	−0.945–−0.142	<0.001	-	-	-
Triglycerides (mg/dL) * 100	1.937	0.461–3.413	<0.05	-	-	-
log_10_ (NT-proBNP (pg/mL))	10.928	8.701–13.156	<0.001	4.416	1.904–6.927	<0.001
eGFR (mL/min/1.73 m^2^)	−0.107	−0.158–−0.055	<0.001	-	-	-
log_10_ (IL-6 (pg/mL))	16.028	12.691–19.365	<0.001	11.022	7.608–14.436	<0.001
log_10_ (hs-CRP (mg/dL))	5.570	3.255–7.884	<0.001	-	-	-
log_10_ (HOMA-IR)	0.809	−2.404–4.021	0.62	-	-	-
log_10_ (ACR)	1.002	−0.533–2.537	0.20	-	-	-
Medication						
β-blockers, *n* (%)	1.980	−0.138–4.099	0.07	-	-	-
ACE-I or ARB, *n* (%)	0.654	−1.419–2.727	0.54	-	-	-
Statins, *n* (%)	−1.436	−3.837–0.966	0.24	-	-	-

Abbreviations: BMI—body mass index, AF—atrial fibrillation, MI—myocardial infarction, HT—heart failure, COPD—chronic obstructive pulmonary disease, IL-6—interleukin 6, hsCRP—high-sensitivity C-reactive protein, HOMAR-IR—homeostatic model assessment, ACR—albumin-to-creatinine ratio, ACE-I—angiotensin-converting-enzyme inhibitor, ARB—angiotensin receptor blockers. * Statistical significance was set at a *p*-value below 0.05.

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
