# Peer review of "CA125 as a Marker of Heart Failure in the Older Women: A Population-Based Analysis"

_jcm, 2019, doi:10.3390/jcm8050607_

Reviewer 1 Report

The study „CA125 as a marker in cardiovascular disease in the older women: population-based analysis” by Bedowska and colleagues evaluates the levels of the potential cardiovascular marker CA125 in elderly women with regards to cardiovascular pathologies.

The paper is well written. Since CA125 seems to be a promising marker in cardiovascular pathologies, the topic is of interest for the readers. However, I have some comments and suggestions regarding the study.

I am worried about the data collection. As already pointed out by the authors in their limitations section, data collection was performed by socioeconomic survey. With respect to an impaired mental status in some elderly patients, I am concerned, that data might be missed or misclassified. This might be even more important regarding the high rate of asymptomatic myocardial infarction in women. Did the authors take any precaution to prevent this issue?

Furthermore, the study design does not allow the evaluation of the severity of the assessed pathologies. Nevertheless, it would be important to correlate the levels of CA125 with NYHA-class, echocardiography data ect.

I would be also interested in a more specific characterization of the pathologies. Of note, heart failure with preserved ejection fraction is an important issue in the elderly population. Are the authors able to provide any data on this issue?

This is not the first trial dealing with this topic. Ma et al. performed a similar trial in 2013. Nevertheless, as already correctly mentioned by the authors their population mainly compromised of males. Furthermore, the levels of CA125 in patients with acute cardiac decompensation were evaluated by Nunez et al.. Of note, the mean age in their study was similar to the age of the presented cohort. They found a prognostic value of the discussed biomarker.

Unfortunately, due to the cross-sectional character of the study the authors are not able to provide any data on the prognostic impact of CA125 in this population. Therefore, I would wish to have a better clarification of the clinical significance of the data in the discussion section.

Minor issues

In the table legends the authors should state, if data is provided as mean and SD.

 Please correct some minor grammar mistakes: line 55, 59, 223 and 259-260

Author Response

Point 1: I am worried about the data collection. As already pointed out by the authors in their limitations section, data collection was performed by socioeconomic survey. With respect to an impaired mental status in some elderly patients, I am concerned, that data might be missed or misclassified. This might be even more important regarding the high rate of asymptomatic myocardial infarction in women. Did the authors take any precaution to prevent this issue?

Response 1:PolSenior study was much more than a socioeconomic survey. As it is described in mentioned in the assumptions and objectives of the PolSenior project (Exp Gerontol. 2011;46(12):1003-9) the study covered medical, psychological and socioeconomic aspects. The study was performed by trained nurses during three visits, and included questionnaire survey, comprehensive geriatric assessment and blood and urine sampling. The questionnaire consisted of medical and specific socioeconomic questions. The comprehensive geriatric assessment included blood pressure and anthropometric measurements, as well as selected scales and tests routinely used in the examination of elderly subjects. All nurses were trained for the purpose of the study. All cardiovascular events collected in the survey were confirmed by hospital discharge carts. However, it  is true that the detection of asymptomatic myocardial infarction, and compensated HF was not possible. We added the information about possible missing data as the limitation of the study.

Point 2: Furthermore, the study design does not allow the evaluation of the severity of the assessed pathologies. Nevertheless, it would be important to correlate the levels of CA125 with NYHA-class, echocardiography data ect.

Response 2: We fully agree with the Reviewer that study design does not allow the evaluation of the severity of HF and analysis of the correlation between CA125 and NYHA-class as well as echocardiography will be very interesting and important. However, the PolSenior project has epidemiological character and the survey does not included detailed clinical data describing HF severity. In our opinion our data suggest indicate the sensibility of undertaking further clinical studies, which will allow to accurately assess the occurrence of such dependencies. Our intention is to conduct such a study and thank you for this valuable consideration. According to this comment we changed the aim of the study so that it is adapted to your data: ‘The main aim of this study was to assess, in the population of the older Polish women, relationships between serum concentration of CA125 and NT-proBNP’.

Point 3: I would be also interested in a more specific characterization of the pathologies. Of note, heart failure with preserved ejection fraction is an important issue in the elderly population. Are the authors able to provide any data on this issue?

Response 3: As was mentioned above due to epidemiological character of the study such data is unavailable, but thank you for this consideration and we will obtain it during the planned study.

Point 4:This is not the first trial dealing with this topic. Ma et al. performed a similar trial in 2013. Nevertheless, as already correctly mentioned by the authors their population mainly compromised of males. Furthermore, the levels of CA125 in patients with acute cardiac decompensation were evaluated by Nunez et al.. Of note, the mean age in their study was similar to the age of the presented cohort. They found a prognostic value of the discussed biomarker.

Response 4: We agree with Reviewer that similar trials were performed, but these studies were performed in Chinese and Spanish not Caucasian race population.

This issue has been clarified: ‘To the best of our knowledge, the current study is the first to investigate association between CA125 and NT-proBNP levels in the old Caucasian female general population. Ma et al. [14] conducted a similar study, but among Chinese patients hospitalized for chronic HF aged 85 and older, with the majority (84%) of the study group being male’.

Despite the similar mean age of patients in other studies, this study is the first that had so wide age range - 65-102 years. It allowed us to assess the reliable relationship between age and Ca125 in the elderly female population.

Point 5: Unfortunately, due to the cross-sectional character of the study the authors are not able to provide any data on the prognostic impact of CA125 in this population. Therefore, I would wish to have a better clarification of the clinical significance of the data in the discussion section.

Response 5: We sought to analyse the prognostic impact of CA125 level, but there is low number of subjects (N=79) with the level over the established cut-off value (>35). A lager cohort would be suitable for such analysis. Discussion was corrected: ‘On the basis our results the assessment of usefulness of CA125 as a marker of severity of HF and mortality due to HF is impossible. However, we suggested that CA125 may be a complementary to NT-proBNP tool to seeking women with high risk of HF which need further diagnostics’.

Point 6: In the table legends the authors should state, if data is provided as mean and SD.

Response 6: The description was corrected [mean± SD or median (1-3Q)].

Point 7: Please correct some minor grammar mistakes: line 55, 59, 223 and 259-260

Response 7: Grammar mistakes were corrected:

- Several recent studies have shown that CA125 serum levels were also elevated in heart failure (HF), atrial fibrillation (AF), ischemic heart disease [6-8].

- It is believed that neurohormonal and inflammatory activation as well as increased central venous volume and congestion are the factors increasing the level of CA125 [11-12].

-Ma et al. [14] conducted a similar study, but among Chinese patients hospitalized for chronic HF aged 85 and older, with the majority (84%) of the study group being men

- Recently published studies have demonstrated the independent effect of permanent AF on CA125 values [29]. However, despite the association between CA125 and IL-6 observed in our study, history of AF did not influence CA125 concentration.

Reviewer 2 Report

The study by Bedkowska et al evaluated the influence of medical history, baseline demographics and some biochemical indices (lipids and some inflammatory parameters) on CA125, a tumor marker, in 1565 women 65-102 years. A history of malignancy was an exclusion criteria. Using a stepwise regression approach they identified that high IL-6 and proBNP, age and a history of HF hospitalization and COPD as independent predictors of CA125 levels.

As noted from the introduction and in the discussion there is literature supporting these associations, e.g. there reports of elevated CA125 in HF as well as data showing it is associated with poor prognosis.

The aims as outlined at the end of the introduction is “The main aim of this study was to assess, in the population of the older Polish women, relationships between serum concentration of CA125 and atrial fibrillation, coronary artery disease, myocardial infarction, past hospitalization for heart failure, serum NT-proBNP levels, IL-6 levels and clinical data.”

Although I believe the data may support some association between CA125 and HF in elderly women, I am no sure the statistical approach is the appropriate one and question the usefulness of these data.

  The authors need to better define the rationale for the study, merely looking at associations     between a marker and demographics does not bring the field forward.

The authors found CA125 and HF and suggest it could be useful in monitoring HF. Do they mean it could be used to identify HF? Monitor the progression of HF? As noted in the discussion, there is data on CA125 in HF as well as prognosis. The authors need to discuss the usefulness of their data in a clinical setting

Author Response

Point 1: Although I believe the data may support some association between CA125 and HF in elderly women, I am no sure the statistical approach is the appropriate one and question the usefulness of these data.

Response 1: According to Reviewer suggestion we modified the aim of the study – ‘ The main aim of this study was to assess, in the population of the older Polish women, relationships between serum concentration of CA125 and NT-proBNP ’.

Concerning the statistical analysis we think, that the linear regression has a wide-spread application since years. Moreover there is no doubt that this is one of the most powerful analysis in statistical biomedical data analysis provided fulfilling of the required assumption. In our case that was not and thus we have decided to use a t-skew regression to assure the proper data analysis and conclusion process. The results are presented in the established manner. Therefore we are convinced that the statistical analysis has been done correctly and according to statistical data analysis guidelines.

Point 2: The authors need to better define the rationale for the study, merely looking at associations between a marker and demographics does not bring the field forward.

Response 2: The rationale was modified according to corrected aim of the study.

Point 3: The authors found CA125 and HF and suggest it could be useful in monitoring HF. Do they mean it could be used to identify HF? Monitor the progression of HF? As noted in the discussion, there is data on CA125 in HF as well as prognosis. The authors need to discuss the usefulness of their data in a clinical setting

Response 3: Discussion was corrected we hope that now meets the requirements of the Reviewer.

‘In summary our study shown relationship between CA 125 and NT-proBNP levels. In addition, we observed that increased CA 125 levels may be related to numerous factors such as age, hospitalization for HF, history of AF and COPD, as well as inflammatory markers (IL-6, hs-CRP). Thus, clinical usefulness of CA125 as a marker of HF severity is limited. However, CA125 may be useful, complementary to NT-proBNP, for the risk assessment of HF.  The usefulness of CA 125 as a prognostic marker for HF severity and mortality in Caucasian female population requires further study.’

Round  2

Reviewer 1 Report

I thank the authors for their revsion. Despite improvements of their work, I have still some important issues regarding their revision:

- Unfortunately, as mentioned by the authors, the study is not able to provide any data on the prognostic value of CA125

- In their revised version, the authors suggest CA125 being a complementary parameter to NT-proBNP. I would suggest to provide a better explanation of their thesis. Due to the descriptive character of their study this conclusion is difficult to make.

- The new paragraphs (80-90) and (418-424) are difficult to understand, the authors should revise their style to clarify their intentions

- The manuscript (especially the new paragraphs) has still to be checked for grammar mistakes

Author Response

Answers to the reviewer comments:

 Point 1 : Unfortunately, as mentioned by the authors, the study is not able to provide any data on the prognostic value of CA125.

Response 1:  It is a statement. The answer  seems not be necessary. 

 Point 2:   In their revised version, the authors suggest CA125 being a complementary parameter to NT-proBNP. I would suggest to provide a better explanation of their thesis. Due to the descriptive character of their study this conclusion is difficult to make.

Response 2: We agree that this statement is not supported by our data. We have change the comclusion to: “Despite the association between CA125 and NT-proBNP, the usefulness of CA125 for the detection of HF in the older women is limited by factors such as inflammatory status and age.”

In line with the change we have modified a part of discussion: “On the basis of our results the assessment of CA125 usefulness as a marker of HF severity and mortality for HF is not possible. The association between CA125 and NT-proBNP levels is not enough to prove its’ usefulness in work-up of HF.”

 Point 3:    The new paragraphs (80-90) and (418-424) are difficult to understand, the authors should revise their style to clarify their intentions.

Response 3: We have modify these paragraphs to clarify our intentions.

Point 4:  The manuscript (especially the new paragraphs) has still to be checked for grammar mistakes.

Response 4: We did our best but we are aware that we are not natives. Therefore, we choose a paid option go get the support from the MPDI editors.